# Genome-Wide Identification and Expression Analysis of *nsLTP* Gene Family in Rapeseed (*Brassica napus*) Reveals Their Critical Roles in Biotic and Abiotic Stress Responses

**DOI:** 10.3390/ijms23158372

**Published:** 2022-07-28

**Authors:** Yufei Xue, Chunyu Zhang, Rui Shan, Xiaorong Li, Alain Tseke Inkabanga, Lejing Li, Huanhuan Jiang, Yourong Chai

**Affiliations:** 1College of Agronomy and Biotechnology, Southwest University, Chongqing 400715, China; xyf710@swu.edu.cn (Y.X.); zcy20010122@email.swu.edu.cn (C.Z.); despacitobei@email.swu.edu.cn (R.S.); lxr1026@email.swu.edu.cn (X.L.); alain1982@email.swu.edu.cn (A.T.I.); lilejing123@email.swu.edu.cn (L.L.); jh8469259@email.swu.edu.cn (H.J.); 2Faculté des Sciences Agronomiques, Université Pédagogique Nationale (UPN), Kinshasa 8815, Democratic Republic of the Congo

**Keywords:** nsLTP, rapeseed (*Brassica napus*), genome-wide identification, phylogenetic analysis, expression analysis

## Abstract

Non-specific lipid transfer proteins (nsLTPs) are small cysteine-rich basic proteins which play essential roles in plant growth, development and abiotic/biotic stress response. However, there is limited information about the *nsLTP* gene (*BnLTP*) family in rapeseed (*Brassica napus*). In this study, 283 *BnLTP* genes were identified in rapeseed, which were distributed randomly in 19 chromosomes of rapeseed. Phylogenetic analysis showed that BnLTP proteins were divided into seven groups. Exon/intron structure and MEME motifs both remained highly conserved in each BnLTP group. Segmental duplication and hybridization of rapeseed’s two sub-genomes mainly contributed to the expansion of the *BnLTP* gene family. Various potential *cis*-elements that respond to plant growth, development, biotic/abiotic stresses, and phytohormone signals existed in *BnLTP* gene promoters. Transcriptome analysis showed that *BnLTP* genes were expressed in various tissues/organs with different levels and were also involved in the response to heat, drought, NaCl, cold, IAA and ABA stresses, as well as the treatment of fungal pathogens (*Sclerotinia sclerotiorum* and *Leptosphaeria maculans*). The qRT-PCR assay validated the results of RNA-seq expression analysis of two top *Sclerotinia*-responsive *BnLTP* genes, *BnLTP129* and *BnLTP161*. Moreover, batches of *BnLTPs* might be regulated by BnTT1 and BnbZIP67 to play roles in the development, metabolism or adaptability of the seed coat and embryo in rapeseed. This work provides an important basis for further functional study of the *BnLTP* genes in rapeseed quality improvement and stress resistance.

## 1. Introduction

Rapeseed (*Brassica napus*), which provides abundant vegetable oil and substantially potential biodiesel feedstocks, is one of the most important oilseed crops worldwide [1]. The growth of the world’s population and the improvement of living standards have increased the global demand for rapeseed oil [2,3]. In addition, the yield and oil quality of rapeseed often suffer from extensive losses, owing to pathogen and biotic/abiotic stresses during rapeseed production worldwide [4,5,6,7]. These situations have increased the demand for the study of the genetic basis of high seed oil accumulation and high stress resistance of rapeseed.

With the development of next-generation sequencing technology, rapeseed breeders have conducted numerous research studies on the molecular mechanism of seed oil biosynthesis and stress resistance in rapeseed using genome-wide association and multiple-omics comparative analyses [8,9,10,11,12] in which various types of candidate genes, including plant non-specific lipid transfer protein (*nsLTP*) genes, were obtained. The nsLTPs that are plant-specific can bind or transfer many classes of hydrophobic molecules in vitro, e.g., fatty acids, fatty acyl-CoA, phospholipids, glycolipids and cutin monomers [13,14,15]. The nsLTPs have been shown to play crucial roles in various processes of plant growth and development, including seed development and quality, seed oil accumulation, seed germination, post-meiotic anther development, cutin formation and cell wall extension [16,17,18,19,20,21,22,23,24,25]. The nsLTPs also play critical roles in plant responses to abiotic stresses, e.g., cold, heat, salt and drought, as well as biotic stresses, e.g., bacterial and fungal pathogens [21,23,26,27,28,29,30,31]. Therefore, a systematic identification of rapeseed nsLTPs is of extreme importance for the creation of transgenic or gene-stacking rapeseed materials with enhanced seed oil quality and resistance to stresses.

Seeds are an important place for fatty acid synthesis and metabolism, so it is meaningful to have a proper understanding of nsLTPs involved in seed development, metabolism and adaptation. TT1 and bZIP67 are two representative transcription factors (TF) regulating seed coat and seed embryo developmental traits, respectively [32,33]. Therefore, in this study, differentially expressed genes (DEGs) of *nsLTP* family were identified from the DEG database of transgenic rapeseeds overexpressing *BnTT1* and *BnbZIP67*, and it is speculated that they may be involved in the development and metabolism of seed coat and seed embryo.

The nsLTPs are small, basic proteins that exist in higher plants, which harbor an N-terminal signal peptide and contain an eight-cysteine motif (8CM, C–Xn–C–Xn–CC–Xn–CXC–Xn–C–Xn–C) [34]. The nsLTPs function in the cell secretory pathway [16]. The nsLTPs are located in the plasma membrane [35,36,37], extracellular space [38], cell wall [39] and cytoplasm [40]. The nsLTPs have been reported to play important roles in various developmental and physiological processes [21,40,41]. In recent years, the number of plant genomes sequenced has exploded, which facilitates genome-wide analysis of the plant *nsLTP* gene family in a variety of plant species. The model dicot Arabidopsis has been reported to contain 79 *nsLTP* genes [42]. Studies showed that there are 77 in rice [42], 63 in maize [42], 58 in sorghum [42], 461 in wheat [43], 70 in barley [44], 63 in *B. rapa* [45], 89 in cabbage [46], 83 in potato [47], 64 in tomato [48], 51 in *Gossypium arboretum* [49], 47 in *G. raimondii* [49], 91 in *G. hirsutum* [49], 52 in sesame [50] and 64 in *Arachis duranensis* [51]. To date, there has been no systemic whole-genome analysis of the *nsLTP* gene (*BnLTPs*) family in *B. napus* under various abiotic/biotic stresses, as well as in seed development and germination, and seed coat color formation, although the reference genomes of different cultivars or ecotypes of *B. napus* have been published. Herein, we systemically identified *BnLTPs* in *B. napus*, and then performed the detailed analyses containing phylogenetic relationship, gene structure, MEME motifs, chromosome location, gene duplication and synteny, SSR loci and *cis*-element in promoters. We also detected their expression patterns in various organs and tissues and in response to biotic/abiotic stresses and phytohormone treatments. This study will lay a solid foundation for further functional research of *BnLTP* genes as well as their application in improvement of crop quality and stress resistance in *B. napus*.

## 2. Results

### 2.1. Identification of BnLTP Genes

A total of 283 *nsLTP* genes were identified in the *B. napus* genome (Appendix A). Gene sequences of *BnLTP* genes varied from 273 bp to 8989 bp in length, and *BnLTP* mRNAs ranged from 92 bp to 1809 bp in length. BnLTP proteins varied from 90 aa to 400 aa in length. BnLTP proteins all contained conserved 8CM domains (Figure 1; Appendix A). Of the 283 BnLTP proteins, 76 harbored C-terminal GPI anchor signal, whereas it is absent in the remaining 207 BnLTP proteins. The theoretical *M*_W_s of BnLTP proteins ranged from 9658.62 Da to 43,916.10 Da. The *pI* values of BnLTP proteins varied from 3.92 to 11.00. There existed 18–34 aa N-terminal signal peptides in all BnLTP proteins, and thus they all could be substrates for the cell’s secretory pathways. In addition, for further evolutionary relationship analysis, a total of 127, 151 and 106 *nsLTP* genes were identified in *B. rapa*, *B. oleracea* and *A. thaliana*, respectively (Appendix A).

### 2.2. Phylogenetic Relationship Analysis

In total, 667 8CM sequences of nsLTP proteins from *B. napus*, *B. rapa*, *B. oleracea* and *A. thaliana* (Appendix A), were utilized to construct the phylogenetic tree. Evolutionary analysis showed that these 667 nsLTP proteins from four species were separated into seven groups: I, II, III, IV, V, VI and VII (Figure 2). As shown in Table 1, these seven groups all existed in three *Brassica* species and *A. thaliana*. In total, there were 170 *nsLTP* genes from four species in group I, 118 in group II, 68 in group III, 46 in group IV, 98 in group V, 80 in group VI, and 87 in group VII. 

### 2.3. Analysis of Gene Structure and Conserved Motifs

To detect exon-intron organizations of *BnLTP* genes, their gene structures were illustrated using the GFF3 file from a *B. napus* genome on the TBtools program (Appendix A). Of 283 *BnLTP* genes, 143 contained no intron (Appendix A). Of the remaining 140 *BnLTP* genes containing introns, 64 contained 1 intron, 62 had 2 introns, 5 possessed 3 introns, 4 harbored 4 introns, 2 included 5 introns, 2 contained 7 introns, and 1 had 7 introns. To identify conserved motifs in BnLTP protein sequences, MEME tools were used to perform the prediction and analysis. As shown in Appendix A, there existed 20 conserved MEME motifs in 283 BnLTP proteins.

### 2.4. Chromosome Location

The physical chromosome locations of *BnLTP* genes were visualized using the TBtools. In total, 239 of 283 *BnLTP* genes were mapped to 19 chromosomes of *B. napus,* while the remaining 44 *BnLTP* genes were mapped to random chromosomes (23 on the A-subgenome, 18 on the C-subgenome, and 3 on chrUnn_random, Appendix A). Notably, these *BnLTP* genes were unevenly distributed to the chromosomes. Chromosomes A09, C02 and C03 all contained 19 *BnLTP* genes, chromosomes C01 and C04 both contained 11, chromosomes A01 and A06 both contained 10, chromosomes A05 and C06 both contained 7 and chromosomesA04 and A07 both contained 6. Chromosomes A03, A02, C09, A08, C08 and C07 contained 26, 18, 16, 15, 13, 12 and 9, respectively. In total, there existed 38 clusters containing 92 *BnLTPs* genes in 15 chromosomes and 4 random chromosomes (Appendix A). Chromosome A09 harboring 13 *BnLTP* genes contained the most clusters (6). Most clusters (27) contained 2 genes, 6 clusters harbored 3 genes and 5 clusters had 4 genes.

### 2.5. Gene Duplication and Synteny Analysis

To reveal the mechanism underlying the expansion of the *nsLTP* gene family in *B*. *napus*, we detected the types of *Bn**LTP* gene duplication. Of 283 *BnLTP* genes, 191 (67.4%) and 31 (10.9%) genes were derived from segmental duplication and tandem duplication (Appendix A). It appears that segmental duplication played important roles in the expansion of the *BnLTP* gene family. Based on the genome information from *B. napus*, we analyzed the synteny relationship of *BnLTP* genes (Figure 3). For the 283 *BnLTP* genes, 204 syntenic gene pairs were found in *B. napus* genome in which 40 (19.6%), 23 (11.2%) and 141 (69.1%) gene pairs were obtained among BnA-BnA subgenomes, BnC-BnC subgenomes and BnA-BnC subgenomes, respectively. Overall, the expansion of the *BnLTP* gene family might have resulted from segmental duplication and hybridization of its two sub-genomes.

### 2.6. Analysis of cis-Regulatory Elements

To identify *cis*-elements of the promoter regions of *BnLTP* genes, “Search for CARE” in PlantCARE database using 1.5 kb region upstream of ATG start codon of each *BnLTP* gene (Appendix A) as a query was performed. The results showed that four types of *cis*-elements were identified in the promoters of *BnLTP* genes (Appendix A). In the first type, plant growth- and development-related *cis*-elements included sequences associated with differentiation of the palisade mesophyll cells (HD-Zip 1), cell cycle regulation (MSA-like), zein metabolism regulation (O2-site), seed-specific regulation (RY-element), meristem expression (CAT-box), root specific (motif I), circadian control (circadian), endosperm expression (GCN4_motif), endosperm-specific negative expression (AACA_motif) and flavonoid biosynthetic genes regulation (MBSI). In the second type, phytohormone responsive *cis*-elements included sequences associated with auxin-responsive (TGA-element, AuxRR-core), gibberellin-responsiveness (TATC-box, GARE-motif, P-box), abscisic acid responsiveness (ABRE), MeJA-responsiveness (CGTCA-motif, TGACG-motif) and salicylic acid responsiveness (SARE, TCA-element). In the third type, abiotic stress-related *cis*-elements included sequences associated with light responsiveness (G-Box, G-box, ACE, 3-AF1 binding site, AAAC-motif, GT1-motif, Sp1, MRE), anoxic specific inducibility (GC-motif), anaerobic induction (ARE), drought-inducibility (MBS), dehydration, low-temp, salt stresses (DRE) and low-temperature responsiveness (LTR). In the fourth type, biotic-stress-related cis-elements included sequences associated with defense and stress responsiveness (TC-rich repeats) and wound responsiveness (WUN-motif).

### 2.7. Analysis of SSR Loci

MISA-web was used to carry out the analysis of SSR loci. In total, 37 SSR loci were located in *BnLTP* genes/promoters (Appendix A), which were divided into 5 types: mono-(3), di-(23), tri-(4), tetra-(4) and complex-(4) nucleotide repeats. These SSR motifs were mainly harbored in promoters (30), and only a few were found in introns (4), 5′-UTRs (2) and 3′-UTR (1).

### 2.8. Expression of nsLTPs in Different B. napus Tissues

To examine the expression patterns of *BnLTP* genes, we analyzed the transcription levels of *BnLTP* genes in 42 various organs/tissues from different development stages, including roots, stems, leaves, buds, anthocauli, sepals, petals, pistils, stamens, anthers, filaments, seeds, funiculus and silique pods using RNA-seq data (Appendix A). Of the 283 *BnLTP* genes, 29 were not expressed at any tissues, whereas the remaining genes were expressed in various tissues with the different levels (Figure 4 and Appendix A), which included organ-specific and constitutive expression patterns. For example, seed-specific genes contained *BnLTP017, BnLTP117* and *BnLTP210*. Of 38 gene clusters, 18 had similar expression patterns among 42 different tissues, and those of the remaining ones were different (Appendix A). Of 74 duplicated gene groups, 50 harbored similar expression patterns across 42 various organs, and 20 had different expression patterns (Appendix A). These results showed that members of some duplicated gene groups (56.8%) or gene clusters (52.6%) with the different expression patterns might have undergone functional divergence.

Totally, transcriptome data of 185 *BnLTP* genes were obtained for eight LMD-obtained tissues from the early globular seed (Appendix A). A total of 50 were expressed in the embryo proper, 90 in the micropylar endosperm, 49 in the peripheral endosperm, 47 in the chalazal endosperm, 36 in the chalazal proliferating tissue, 56 in the chalazal seed coat, 66 in the inner seed coat and 64 in the outer seed coat (FPKM values > 1). In addition, each tissue contained a unique set of *BnLTP* transcripts. For example, the embryo-specific gene set contained 9 *BnLTP* genes, and that of micropylar endosperm harbored 34. This result showed that some *BnLTP* genes might play roles in tissue-specific development processes of the globular seed.

In total, RNA-seq data of 143 *BnLTP* genes were acquired for three LMD-acquired tissue types of the funiculus (Appendix A). A total of 82, 83 and 58 *BnLTP* genes were expressed in epidermis, cortex and vasculature (FPKM values > 1), respectively, of which 11, 27 and 10 were tissue-specific in corresponding tissues. This result showed the possible crucial roles of *BnLTP* genes in tissue-specific development processes of the *B. napus* funiculus.

### 2.9. Expression of nsLTPs in Response to Biotic Stresses

Plant nsLTPs have been listed as pathogenesis-related protein-14 family [52], and they have been reported to be involved in plant disease resistance reactions [53,54]. Blackleg disease (caused by *L. maculans*) and white stem rot (caused by *S. sclerotiorum*) are the most severe diseases of *B. napus*. We thus detected the expression response of *BnLTPs* to these two fungal diseases.

In response to *S. sclerotiorum*, except for *BnLTP016*, *BnLTP031*, *BnLTP252* and *BnLTP265*, the transcriptome data of the remaining 279 *BnLTP* genes were obtained (Figure 5; Appendix A), of which 152 were expressed (FPKM values > 1). A total of 72 *BnLTP* genes were *S. sclerotiorum*-responsive genes (*p* < 0.05) of which *BnLTP033*, *BnLTP129*, *BnLTP161* and *BnLTP264*, might have more critical roles in resistance to white stem rot. In response to *L. maculans*, except for *BnLTP016*, RNA-seq data of the remaining 282 *BnLTP* genes were obtained (Figure 6; Appendix A) of which 174 were expressed (FPKM values > 1). A total of 96 *BnLTP* genes were *L. maculans*-responsive genes (*p* < 0.05) of which *BnLTP161* and *BnLTP015* might play more important roles in plant defense to blackleg disease.

### 2.10. Expression of nsLTPs in Response to Abiotic Stresses

RNA-seq data of 283 *BnLTP* genes for heat, drought, NaCl, cold and dehydration stresses were all obtained (Figure 7; Appendix A), and there were 65, 74, 76, 80 and 67 expressed in various samples from the treatments (FPKM values > 1), respectively. We found that the numbers of *BnLTP* genes (*p* < 0.05) responding to heat, drought, NaCl, cold and dehydration were 12, 26, 61, 41 and 32, respectively. The responsive mechanisms were different between different low-temperature conditions. Then, we also examined the expression levels of *BnLTP* genes in two early-maturing *B. napus* varieties with different cold tolerance under cold accumulations at chilling (CA) and freezing (FA) temperature and cold shocks at the same temperature (CB and FB) condition. A total of 94, 94, 84 and 90 *BnLTP* genes were expressed under CA, FA, CB and FB treatments, respectively (Appendix A); 24, 26, 39 and 48 were CA-, FA-, CB- and FB-responsive genes, respectively.

Auxin-responsive elements (TGA-element and AuxRR-core) and abscisic acid-responsive elements (ABRE) were found in the promoters of *BnLTP* genes, and thus we analyzed their expression patterns under IAA and ABA treatments. Of the 283 *BnLTP* genes, transcriptome data of 149 and 176 were obtained for IAA and ABA treatment, respectively (Figure 7; Appendix A). We found that 13 *BnLTP* genes were IAA-responsive genes, whereas 22 were ABA-responsive genes (*p* < 0.05).

### 2.11. Expression of nsLTPs in B. napus Seed Germination

To explore the roles of *BnLTP* genes in seed germination trait, we examined their expression patterns during different stages of seed germination of three selected accessions with representative germination rates (high, medium or low; C129, C033 and C032, respectively). During 4 time points, 2, 12, 36 and 72 h after imbibition in these 3 accessions, the numbers of expressed *BnLTP* genes (FPKM values > 1) increased gradually, ranging from 26 to 135 and those of responsive *BnLTP* genes also increased gradually ranging from 1 to 46 (Figure 8 and Appendix A). This result showed that some *BnLTP* genes might be important in different seed germination processes.

### 2.12. Expression of nsLTPs in B. napus Brown Seeds and Yellow Mutation

Of the 55,637 unigenes identified in 26 d brown seeds and yellow mutation, 64 *BnLTP* genes were detected (Appendix A). Only three *BnLTP* genes, *BnLTP089*, *BnLTP110* and *BnLTP257* were found to be DEGs between the materials, which were all up-regulated in yellow mutation. This result showed that they might play important roles in regulating or co-regulating the seed coat color of *B. napus*.

### 2.13. Expression of nsLTPs in B. napus Overexpressing BnTT1 and BnbZIP67

In total, there existed 40, 30 and 32 *BnLTP* family DEGs in the mid-stage seeds of transgenic rapeseeds overexpressing *pBAN::BnTT1* (BOE), *pNapA::BnTT1* (NOE) and *pNapA::BnbZIP67* (B67O), respectively (Figure 9a; Appendix A). Some 15 *BnLTP* DEGs were up-regulated in BOE, 11 in NOE, and 16 in B67O, and meanwhile 25 *BnLTP* DEGs were down-regulated in BOE, 19 in NOE, and 16 in B67O. In addition, we found that two *BnLTP* DEGs both were up-regulated in BOE and NOE, and nine were all down-regulated in BOE and NOE, while five were up-regulated in NOE, and they were down-regulated in BOE (Figure 9b). These *BnLTP* genes might be regulated by *BnTT1* or *BnbZIP67* and play roles in seed development, secondary metabolism, oil accumulation and stress resistance during seed growth and maturation.

### 2.14. Validation of Selected Sclerotinia-Responsive Genes

To validate the results of RNA-seq expression analysis, two top *Sclerotinia*-responsive *BnLTP* genes, *BnLTP129* and *BnLTP161*, were selected to perform the qRT-PCR analysis. Meanwhile, to confirm the availability and feasibility of the plant phenotype imager in the detection of *Sclerotinia* disease, the chlorophyll fluorescence images of seedling leaves under the condition of *Sclerotinia* infection were analyzed. At 30 h and 40 h post-infection (hpi), the expression levels of *BnLTP129* and *BnLTP161* and NPQ values increased compared to that of 20 h post-infection (hpi), while Φ_PSII_ and Fv/Fm values decreased (Figure 10). Under *Sclerotinia* infection, *BnLTP129* and *BnLTP161* expression levels both were up-regulated, and the fluorescence parameters varied (Figure 10). This result validated the accuracy of RNA-seq expression data and also showed the application potential of the plant phenotype imager in the detection of *Sclerotinia* disease in *B. napus*.

## 3. Discussion

The nsLTP proteins belong to the prolamin superfamily with conserved cysteine residues, low molecular mass and a high content of α-helices, which are only found in plants [34,55,56]. The nsLTPs have a hydrophobic cavity that can bind and stabilize various lipid molecules outside membranes [16,17,23,27,57]. An increasing number of studies have shown that plant *nsLTP* genes play an important role in the response to biotic and abiotic stresses and in plant growth and development [11,18,21,23,25,28,29,31]. To date, with the availability and feasibility of plant reference genomes, the *nsLTP* gene family has been identified at the genome-wide level in various plant species, including Arabidopsis, *Gossypium*, sesame, cotton, wheat, potato, tomato, tobacco, barley, maize, cabbage and Chinese cabbage [42,43,44,45,46,47,48,49,50,58,59,60,61]. In this study, 283, 127, 151 and 106 *nsLTP* genes were identified in *B. napus*, *B. rapa*, *B. oleracea* and *A. thaliana* genomes, respectively. It appears that the total number of *nsLTP* genes in allotetraploid *B. napus* was almost the sum of those in its two diploid parent species, *B. oleracea* and *B. rapa*, and three *Brassica* species contained a higher number of *nsLTP* genes than that of *A. thaliana*. This result showed that whole genome triplication of ancestor *Brassiceae* and allopolyploidization between its two progenitors might have contributed to the expansion of the *BnLTP* gene family in *B. napus*. In addition, gene duplication and synteny analysis also further revealed that *BnLTP* gene expansion was mainly derived from segmental duplication and hybridization of its two sub-genomes, which was also consistent with the previous research results of genome-wide analysis of other gene families in *B. napus* [62,63].

Numerous studies have showed that some plant nsLTPs were antimicrobial peptides and belonged to the pathogenesis-related protein (PR-14) family members, which possessed extensive antibacterial and antifungal activities [28,29,64]. It has been reported that *nsLTP* genes played an important role in plant defense to fungal and bacterial pathogens [23,28,29,65]. To date, diverse *nsLTPs* have been applied in generating plant disease resistance using biotechnological means. For example, overexpression and knockdown of *StLTP10* in potato indicated that *StLTP10* positively regulated plant resistance to *Phytophthora infestans* [66]. Cotton *GhnsLTPsA10* positively regulated *Verticillium* wilt and *Fusarium* wilt resistance in transgenic Arabidopsis and VIGS cotton [11]. Overexpression of *AtLTP4.4* in transgenic wheat significantly inhibited the growth of *F. graminearum* in Bobwhite and RB07 lines in the greenhouse and reduced the size of fungal lesions in in vitro leaf assays [67]. Transgenic Arabidopsis expressing wheat *TdLTP4* gene enhanced fungal resistance against *Alternaria solani* and *Botrytis cinerea* [68]. Transgenic wheat lines overexpressing *TaLTP5* exhibited significantly enhanced resistance to both common root rot (*Cochliobolus sativus*) and *Fusarium* head blight (*Fusarium graminearum*) compared to the untransformed control [69]. Overexpression of the pepper *CALTPI* gene in Arabidopsis enhanced the resistance against infection by *Pseudomonas syringae* pv. tomato and *Botrytis cinerea* [70]. Overexpression of *OsLTP1* gene in *B. napus* enhanced its resistance to *S. sclerotiorum* [71]. Overexpression of an *nsLTPs-like* antimicrobial protein gene (*LJAMP2*) from motherwort (*Leonurus japonicus*) enhanced resistance to *S. sclerotiorum* in *B. napus*, whereas its overexpression in *Populus tomentosa* also enhanced resistance to fungal pathogens [65,72]. In this study, in response to two fungal pathogens, we identified 72 *S. sclerotiorum*-responsive *BnLTP* genes and 96 *L. maculans*-responsive *BnLTP* genes. For white stem rot resistance, *BnLTP033*, *BnLTP129*, *BnLTP161* and *BnLTP264*, had more critical roles. For blackleg disease resistance, *BnLTP161* and *BnLTP015* played more roles. Therefore, these *BnLTPs* could be candidates used to generate the wide broad plant disease resistance.

Many studies have showed that members of the plant nsLTP family played important roles in response to abiotic stress as well as plant hormone signals [68,73,74], which could help plants adapt to changes of environment conditions. For example, a lipid transfer protein variant with a mutant eight-cysteine motif causes photoperiod- and thermo-sensitive dwarfsm in rice [25]. The transgenic plants expressing the pepper *CALTPI* gene indicated high levels of tolerance to NaCl and drought stresses at various vegetative growth stages [70]. Overexpression of wheat *TdLTP4* in Arabidopsis contributed to plant growth under various stress conditions including NaCl, ABA, JA and H_2_O_2_ treatments [68]. Previous studies showed that *OsLTPL159* was involved in cold tolerance at the early seedling stage in rice, which could be a candidate allele used to improve rice cold tolerance [30]. Overexpression of *LTP3* in Arabidopsis conferred enhanced freezing tolerance, whereas loss-of-function and overexpression of *LTP3* in Arabidopsis both showed its positive regulatory roles in drought tolerance [75]. Drought-Induced *LTP* (*OsDIL*) was mainly responsive to abiotic stresses, including drought, cold, NaCl, and stress-related plant hormone ABA [73]. Transgenic rice plants overexpressing *OsDIL* were more tolerant to drought stress during vegetative development than wild type. The overexpression of *NtLTP4* in *Nicotiana tabacum* increased the resistance to salt and drought stresses [38]. Here, of the 283 *BnLTP* genes, 12 were heat-responsive genes, 26 were drought-responsive genes, 61 were NaCl-responsive genes, 41 were cold-responsive genes, 32 were dehydration-responsive genes, 13 were IAA-responsive genes and 22 were ABA-responsive genes. It could be speculated that these *BnLTP* members played critical roles in response to various abiotic stressors, which also might be applied in molecular breeding of high abiotic stress resistance in *B. napus*.

Numerous studies have showed that plant *nsLTPs* played diverse roles in various biological processes, including seed development and maturation as well as seed germinations [21,22,23,50]. Overexpression of *SiLTPI.23* in Arabidopsis significantly increased seed oil contents by 17–29% compared with the wild type control [50]. *OsLTPL36* was reported to play an essential role in seed development and germination in rice [22]. In *A. thaliana*, the GPI-anchored lipid transfer proteins acted important roles in the development of seed coats and pollen [76]. In this study, we showed that three *BnLTP* genes, *BnLTP089*, *BnLTP110* and *BnLTP257*, were DEGs between 26 d brown seeds and yellow mutants. In yellow mutants, their expression levels were all up-regulated, indicating potential roles of these three *BnLTP* genes in regulating seed coat color. In addition, based on RNA-seq data of 42 various organs/tissues, we found that several *BnLTP* genes, including *BnLTP059* and *BnLTP085*, displayed seed-specific expression genes, revealing their possible roles in regulating seed development. Each of eight LMD-obtained tissues from the early globular seed possessed a unique set of *BnLTP* transcripts, which also might show their important roles in tissue-specific development processes of the globular seed. In the seed germination process of three selected accessions, numbers of expressed and responsive *BnLTP* genes both gradually increased. Meanwhile, overall, at four time points, high germination rate accession material had more expressed and responsive *BnLTP* genes. This result showed that the expression of these *BnLTP* genes might result in high germination rate.

In addition, there existed 30–40 *BnLTP* DEGs in BOE, NOE and B67O, each of which contained the up-regulated and down-regulated *BnLTP* genes. TT1 and bZIP67 are two critical TFs that regulate seed coat and seed embryo development and metabolism, respectively [32,33]. These *BnLTP* DEGs might play important roles in the development, metabolism or adaptability of the seed coat and embryo in rapeseed by responding to BnTT1 and BnbZIP67. Meanwhile, it found that the *BnLTP* family DEGs might have undergone obvious divergence in regulating physiological functions such as seed development, metabolism and adaptation.

Many studies reported that nsLTPs were defined as small basic proteins [16,17,21,23], but some of the nsLTPs identified in this study, were 400 aa in length, which were atypical nsLTP proteins. This study is not the first attempt to classify nsLTPs. Other researchers attempting a systematic analysis of nsLTPs also found these atypical proteins. For example, some of the nsLTPs in *A. duranensis* were 244 aa in length [51], and some in cabbage were 258 aa in length [46]. Therefore, nsLTPs need to be redefined after the functions of these atypical nsLTPs were validated using biochemistry and molecular biology experiments, and we also believe that the nsLTP community will eventually resolve this conflict in that time. Additionally, it has been shown that some nsLTPs can bind or transfer lipids [13,14,15], and thus this function of BnLTPs will be studied in our future research program.

## 4. Materials and Methods

### 4.1. Sequence Retrieval and Structural Analysis

To identify *BnLTP* genes, the HMM files of nsLTP domain (PF14368 and PF00234) were downloaded from Pfam database (http://Pfam.sanger.ac.uk/; accessed on 12 March 2020), and then the hmmsearch program (e-value 10e^−5^; http://hmmer.org; accessed on 12 March 2020) was used to identify BnLTP proteins against *B. napus* genome v5.0 (http://www.genoscope.cns.fr/brassicanapus/; accessed on 12 December 2019). All candidates were detected using SUPERFAMILY 2 database (https://beta.supfam.org/; accessed on 12 March 2020), sequences examined as non-nsLTP proteins, including proteinase/alpha-amylase inhibitor and seed storage family, 2S albumin, were deleted. The 8CM motifs were manually checked in the remaining sequences, and N-terminal signal peptide was also detected using SignalP-5.0 (http://www.cbs.dtu.dk/services/SignalP/; accessed on 13 March 2020). As results, putative BnLTPs lacking 8CM motifs or N-terminal signal peptide were also excluded. Subsequently, PredGPI (http://gpcr2.biocomp.unibo.it/predgpi/; accessed on 25 March 2020) and TargetP-2.0 (http://www.cbs.dtu.dk/services/TargetP/; accessed on 13 March 2020) were used to predict the presence of C-terminal GPI anchor signal and subcellar location of BnLTP proteins, respectively. To further perform the comparative phylogenetic analysis, *AtLTP*, *BrLTP* and *BoLTP* genes were also identified against Arabidopsis genome v11 (https://www.arabidopsis.org/; accessed on 12 May 2020), *B. rapa* genome v3.0 (http://brassicadb.cn/; accessed on 29 April 2020), and *B. oleracea* genome (http://plants.ensembl.org/; accessed on 12 May 2020) using the above method, respectively. The nsLTPs in these four species were named according to their chromosome position. The ProtParam tool (https://web.expasy.org/protparam/; accessed on 15 April 2020) were used to compute the theoretical isoelectric point (pI) and molecular weight (*M*_W_) of BnLTPs.

### 4.2. Phylogenetic Relationship Analysis

The nsLTP protein sequences from *B. napus*, *B. rapa*, *B. oleracea* and *A. thaliana* were multi-aligned using the MAFFT7 program with default parameters [77], and then the neighbor-joining (NJ) phylogenetic tree was generated using MEGA7 software [78] with the 1000 bootstrap replicates and *p*-distance model. Sequence logo of 8CM motifs of BnLTP family proteins was analyzed using WebLogo (http://weblogo.threeplusone.com/; accessed on 16 June 2020).

### 4.3. Analysis of Gene Structure and Conserved Motifs

Gene exon–intron structure information of *BnLTP* genes was retrieved from the GFF3 file in the *B. napus* genome v5.0, and conserved motif files of BnLTP proteins were generated on MEME suite 5.1.0 (http://meme-suite.org/tools/meme; accessed on 18 June 2020) with default parameters. Finally, the combined image was visualized with TBtools [79].

### 4.4. Chromosome Location, Gene Duplication and Synteny Analysis

The chromosome location information of *BnLTP* genes was obtained from the GFF3 file in *B. napus* genome v5.0, and the image was visualized with the TBtools [79]. *BnLTP* gene clusters were identified if two or more *BnLTP* genes were clustered together on the chromosome, and meanwhile they were separated by no more than three genes. Gene duplication patterns were classified with MCScanX with default parameters. Duplicate gene pairs were determined if the identity of their coding regions was >85%, and they were clustered together in an evolutionary guide tree generated using Clustal Omega tool (https://www.ebi.ac.uk/Tools/msa/clustalo/; accessed on 9 June 2020), which were also annotated as duplicated gene groups. The syntenic relationships of duplicate gene pairs were visualized using Circos (http://circos.ca/software/; accessed on 29 July 2021).

### 4.5. Promoter cis-Element Analysis

The 1.5 kb regions upstream of *BnLTP* coding sequences were obtained from the *B. napus* genome v5.0, and then *cis*-regulatory elements were predicted on PlantCARE database (http://bioinformatics.psb.ugent.be/webtools/plantcare/html/; accessed on 18 January 2021).

### 4.6. Analysis of SSR Loci

SSR loci in *BnLTP* genes or their promoters were checked on MISA-web [80], with default parameters.

### 4.7. Expression Analyses Based on Transcriptome Data

Transcriptome data of *B. napus* in 42 different tissues/organs (project ID PRJNA358784), 9 laser-microdissected tissues from the early globular seed (GSE120360) and 3 laser-microdissected tissues from the funiculus (GSE71859), and in response to two fungal pathogens (*Sclerotinia sclerotiorum*, 24 hpi, GSE81545; *Leptosphaeria maculans*, 0, 3, 7 and 11 days post inoculation, GSE777230), heat (40 °C, 3 h) and drought (withdrawing water, 3 days) (GSE156029), cold (4 °C, 12h) and freezing (−4 °C, 12h) (GSE129220), and IAA treatment (GSE105889; 1 μM IAA, 3 h) as well as in brown seeds and yellow mutation (GSE69137) and seed germination progress (GSE137230; 0, 12, 24, 48, and 72 h after imbibition) were downloaded from the NCBI database (https://www.ncbi.nlm.nih.gov/; accessed on 17 July 2021). Transcriptome data of *B. napus* under dehydration (1 h and 8 h), salt (200 mM NaCl; 4 h and 24 h), ABA (25 uM ABA; 4 h and 24 h) and cold (4 °C, 4 h and 24 h) treatments were downloaded in NGDC (https://ngdc.cncb.ac.cn/; accessed on 23 July 2021; project ID CRA001775). For two or three biological duplicates of each sample, all analyses were conducted using the average FPKM or TPM values of transcripts. *BnLTP* genes were considered as differentially expressed genes or stress-responsive genes if |log_2_(foldchange)| of transcript values of gene expression were > 2 and the FPKM/TPM values > 1. Expression analyses were carried out using DESeq2 R package, and heatmaps were visualized using ClustVis web tool (https://biit.cs.ut.ee/clustvis/; accessed on 23 July 2021) and TBtools [79]. Our group has obtained transgenic rapeseeds overexpressing *BnTT1* and *BnbZIP67* and performed RNA-seq analysis. In this study, their DEGs database was used to analyze the expression patterns of *BnLTP* family genes.

### 4.8. RNA Extraction and qRT-PCR Analysis

Total RNA was isolated from seedling leaves of a *Sclerotinia*-resistant rapeseed cultivar ZS10 at 20 h, 30 h and 40 h after *S. sclerotiorum* (1980) inoculation, respectively, using an RNAsimple Total RNA Kit (DP419, Tiangen, Beijing, China). First-strand total cDNA was synthesized with 1 μg of total RNA using the PrimeScript Reagent Kit with gDNA Eraser (Takara, Dalian, China). The qRT-PCR assay was performed using FastStart Universal SYBR Green Master (Roche, Mannheim, Germany) on BIO-RAD CFX96 Real-time PCR System (Bio-Rad, Irvine, CA, USA). Three replicates were carried out for each reaction. Two top *Sclerotinia*-responsive *BnLTP* genes, *BnLTP129* and *BnLTP161*, Appendix A were selected to perform the qRT-PCR experiment, and the *B. napus ACT7* gene was used as internal control [81].

### 4.9. Chlorophyll Fluorescence Imaging

The plant phenotype imager (device no. 20A00005; a chlorophyll fluorescence imaging system FluorCam7.0; Photon Systems Instruments, Brno, Czech Republic) was used to analyze the fluorescence parameters (Fv/Fm, NPQ and ΦPSII) and chlorophyll fluorescence images of the rapeseed seedling leaves under 20 h, 30 h and 40 h post *Sclerotinia* infection [82].

## 5. Conclusions

In the present study, we performed comprehensive whole-genome mining and analysis of *BnLTP* family genes in rapeseed, including phylogenetic relationship, exon/intron structure, MEME motifs, synteny analysis, SSR loci, promoter *cis*-elements and expression patterns. Our work showed critical roles of the *BnLTP* family genes in plant growth and development, as well as in response to biotic/abiotic stresses and phytohormone treatments. This study will lay a solid foundation for further functional research of *BnLTP* genes in quality improvement and stress resistance.

## Figures and Tables

**Figure 1 ijms-23-08372-f001:**
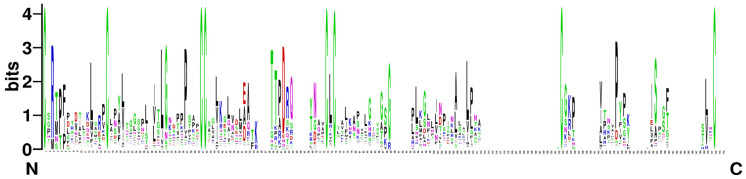
Sequence logos for the eight-cysteine motif (8CM) of BnLTP proteins. The height of each amino acid residue represents the degree of conservation. The numbers on the *x*-axis represent the positions in the 8CM. The *y*-axis shows the information content measured in bits.

**Figure 2 ijms-23-08372-f002:**
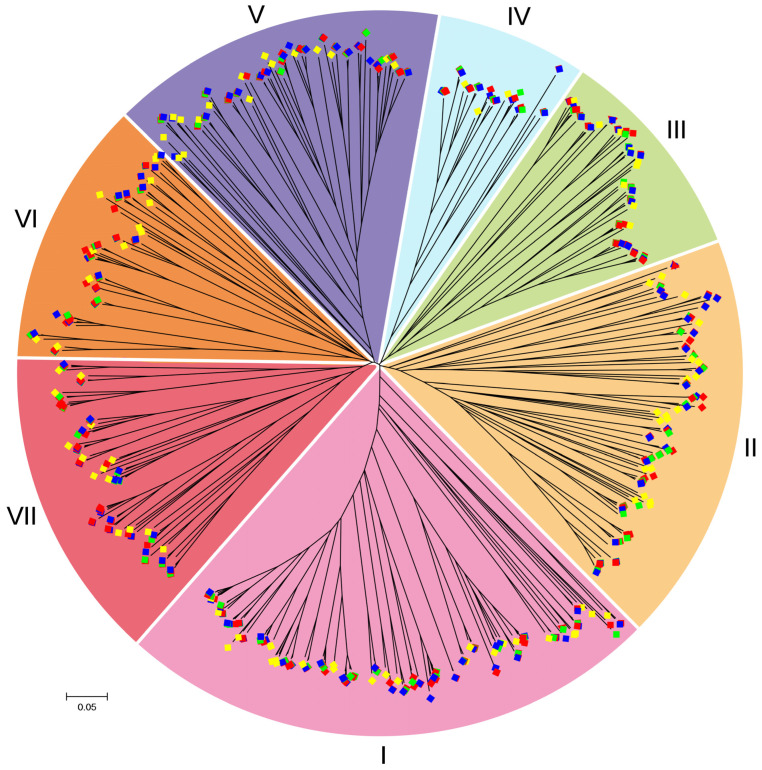
Phylogenetic relationships of nsLTP proteins among *B. napus*, *B. rapa*, *B. oleracea* and *A. thaliana*. This tree was generated using the protein sequences of their 8CM domains. BnLTP, BrLTP, BoLTP and AtLTP are shown with the red (Bn), green (Br), blue (Bo) and yellow (At) squares, respectively.

**Figure 3 ijms-23-08372-f003:**
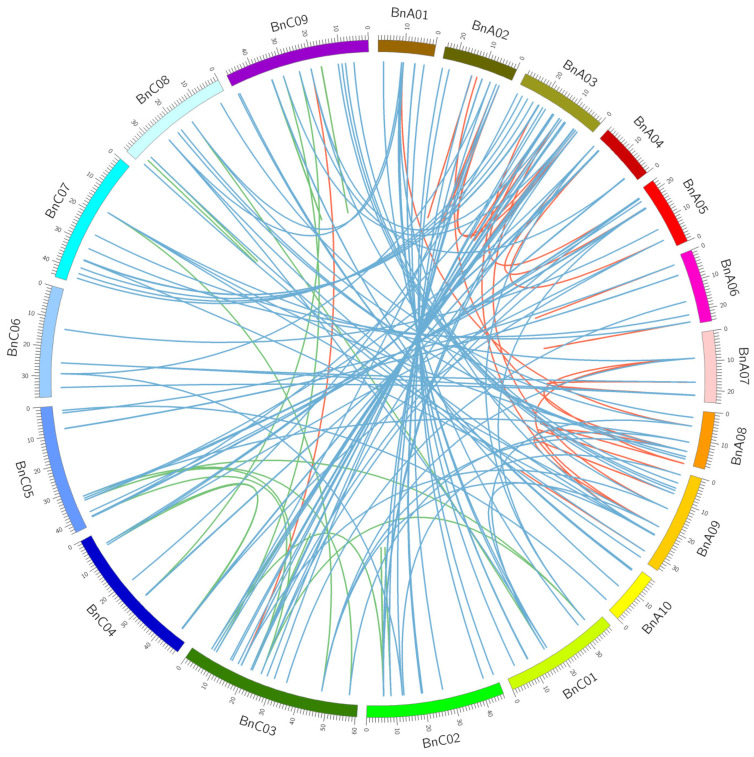
Genome-wide synteny analysis for *nsLTP* genes in *B. napus*. BnA01–10 and BnC01–09 represented chromosomes in A and C sub-genomes in *B. napus*, respectively. Red, green and blue lines linked the syntenic gene pairs within BnA-BnA, BnC-BnC and BnA-BnC subgenomes of *B. napus*, respectively.

**Figure 4 ijms-23-08372-f004:**
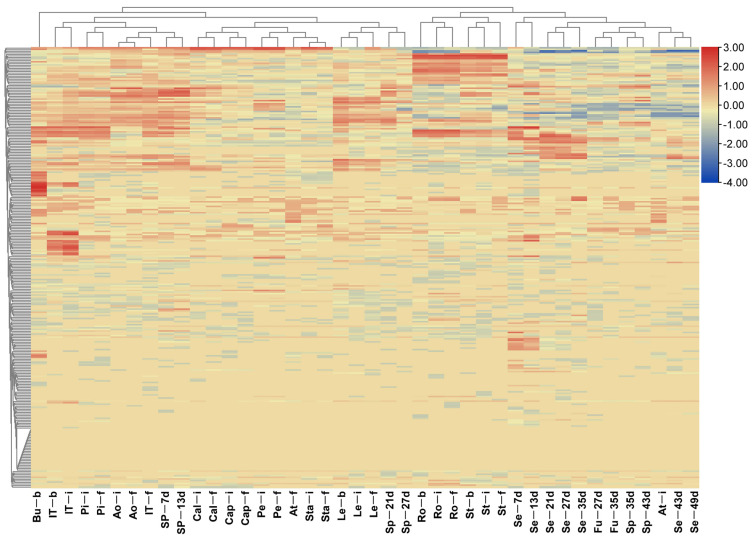
Expression patterns of *BnLTP* genes in 42 various tissues and organs. The abbreviations combinations of 42 samples of *B. napus* cultivar ZS11 are listed in Appendix A. The corresponding heatmap containing *BnLTP* gene names is shown in Appendix A.

**Figure 5 ijms-23-08372-f005:**
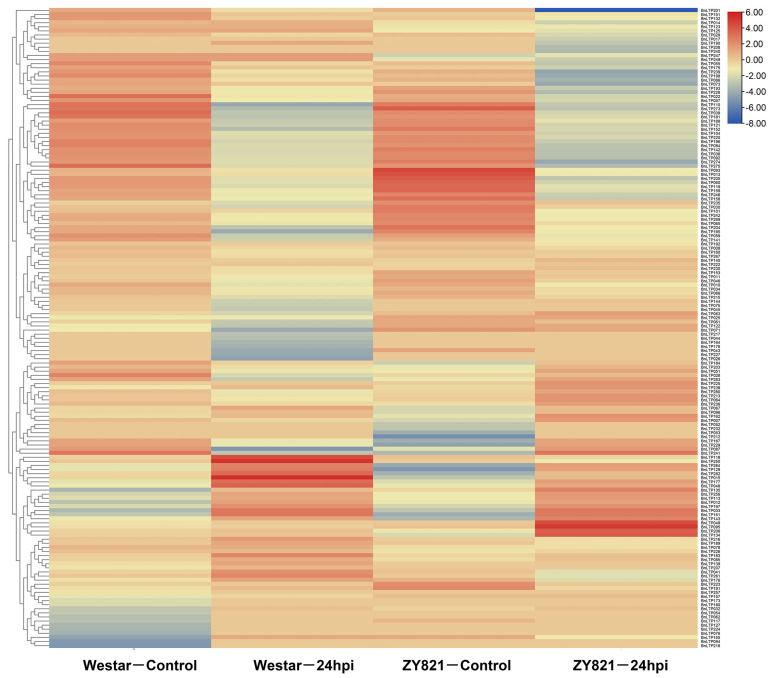
Expression patterns of *BnLTP* family genes in the leaves of susceptible (Westar) and tolerant (ZY821) genotypes of rapeseed infected with *S. sclerotiorum* at 24 h post-inoculation (24 hpi).

**Figure 6 ijms-23-08372-f006:**
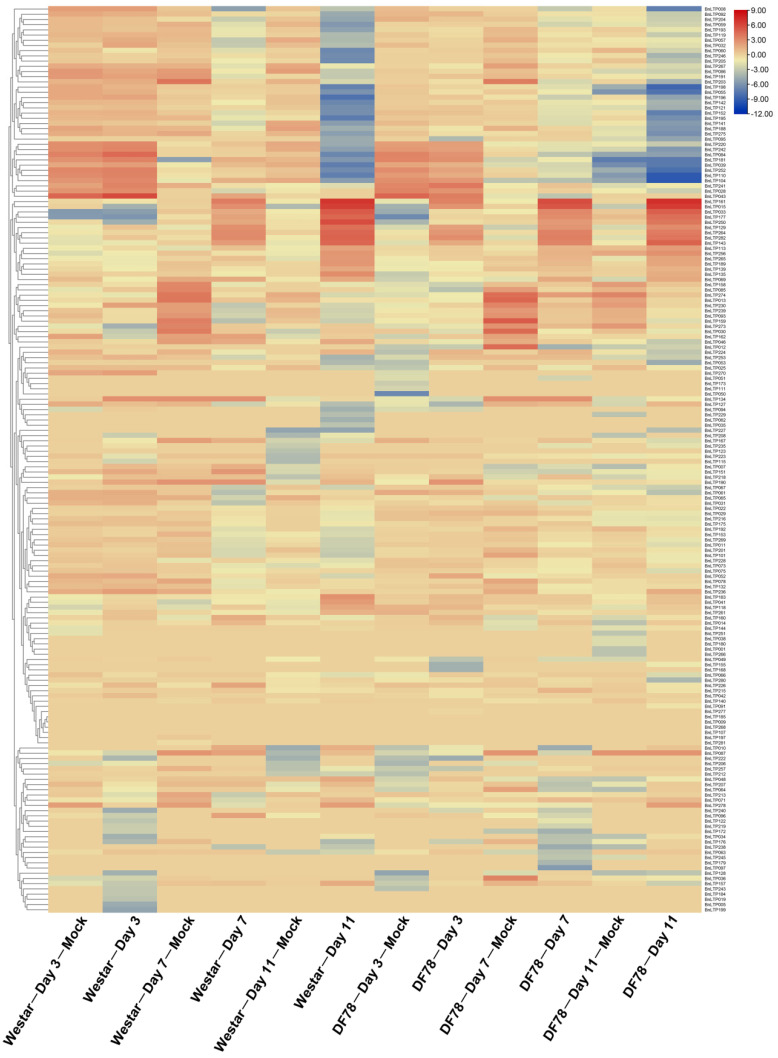
Expression patterns of *BnLTP* family genes in the cotyledons of rapeseed resistant (DF78) and susceptible (Westar) lines at 0, 3, 7 and 11 days post-*L. maculans* inoculation.

**Figure 7 ijms-23-08372-f007:**
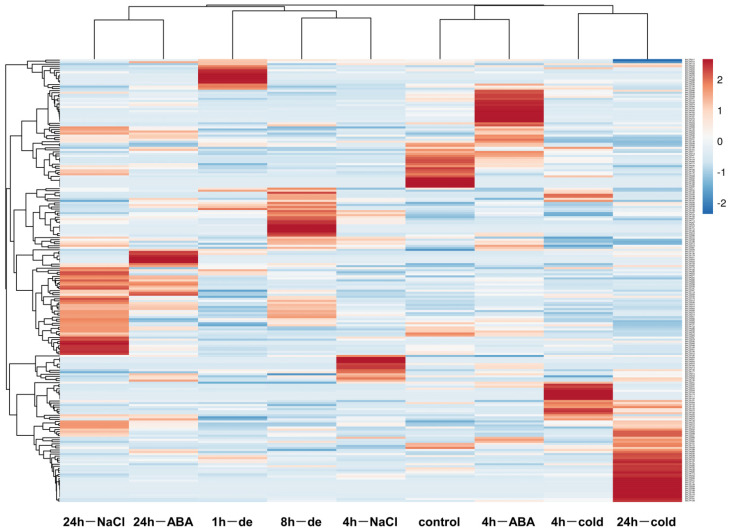
The heatmap of the expression levels of *BnLTP* family genes in 3 week-old plants of rapeseed under multiple abiotic stresses (NGDC project ID CRA001775), including dehydration (1 h and 8 h), ABA (25 μM; 4 h and 24 h), NaCl (200 mM; 4 h and 24 h), and cold (4 °C, 4 h and 24 h).

**Figure 8 ijms-23-08372-f008:**
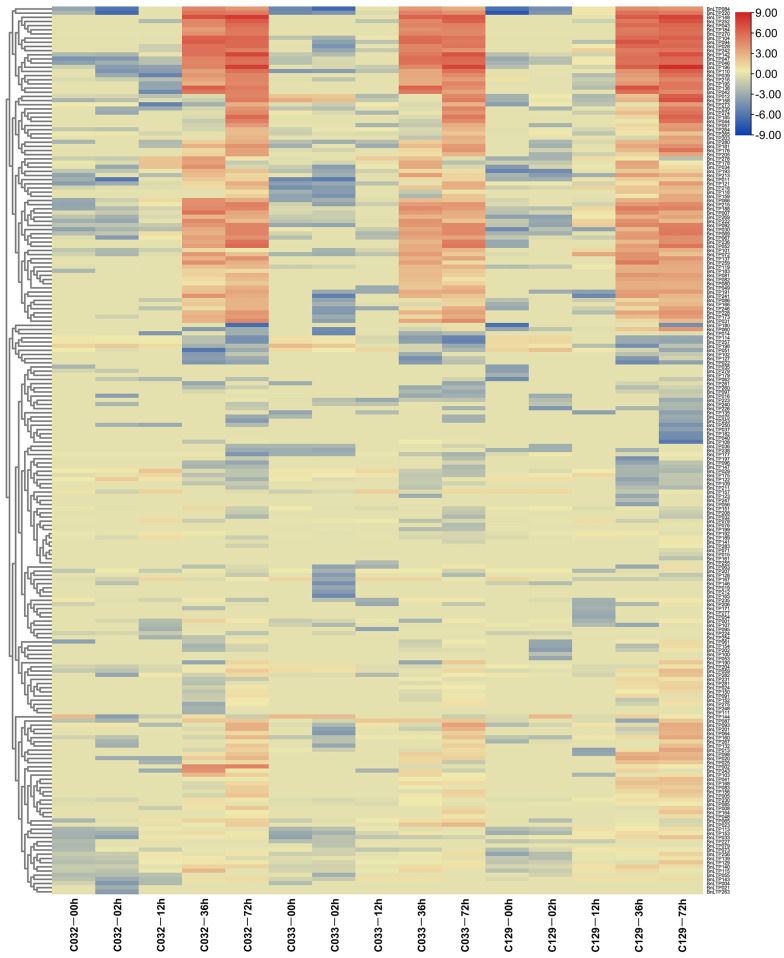
The heatmap of the expression levels of *BnLTP* genes at different germination stages of *B. napus* accessions seeds with representative germination rates (high, medium, or low; C129, C033 and C032, respectively) at 0, 12, 24, 48 and 72 h after imbibition.

**Figure 9 ijms-23-08372-f009:**
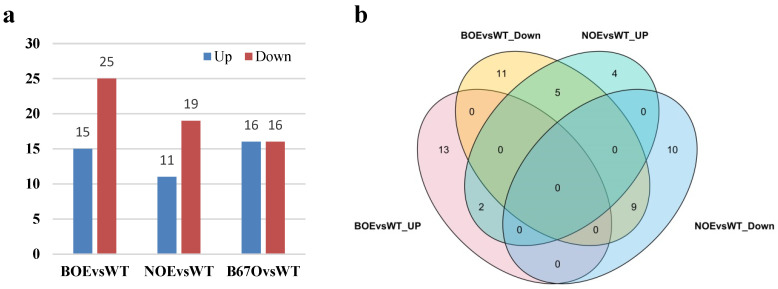
The numbers of *BnLTP* family DEGs in transgenic rapeseeds (Westar cultivar) overexpressing *pBAN::BnTT1* (BOE, 20 DAP seeds, seed coat-specific promoter), *pNapA::BnTT1* (NOE, 20 DAP seeds, seed embryo-specific promoter) and *pNapA::BnbZIP67* (B67O, 25 DAP seeds, seed embryo-specific promoter) (**a**); Venn diagram of the numbers of *BnLTP* family DEGs in BOE and NOE (**b**).

**Figure 10 ijms-23-08372-f010:**
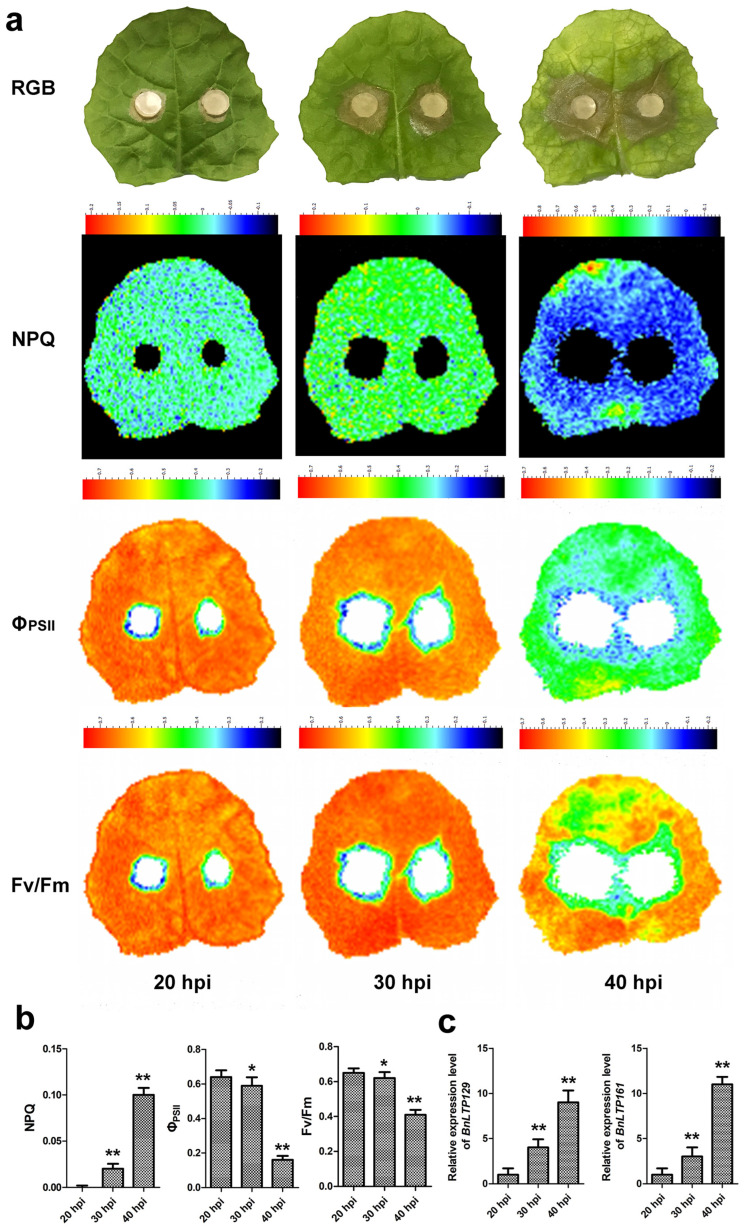
The impact of *S. sclerotiorum* infection on photosynthesis of rapeseed leaves (**a**,**b**); and the qRT-PCR expression level of two top *Sclerotinia*-responsive *BnLTP* genes, *BnLTP129* and *BnLTP161* (**c**). Standard images of the RGB, Fv/Fm, ΦPSII and NPQ from *S. sclerotiorum*-infected rapeseed leaves at 20, 30 and 40 hpi. A false color scale is used for each parameter. The values represent the average ± SD of 3 biological replicates. Fv/Fm, maximum quantum yield of PSII; ΦPSII, effective quantum yield of PSII; NPQ, non-photochemical quenching. * *p* < 0.05 and ** *p* < 0.01 compared with samples at 20 hpi.

**Table 1 ijms-23-08372-t001:** The total number of *nsLTP* genes in each group of *B. napus*, *B. rapa*, *B. oleracea* and *A. thaliana*.

Category	*B. napus*	*B. rapa*	*B. oleracea*	*A. thaliana*
Total	283	127	151	106
I	65	38	40	27
II	49	22	25	22
III	33	11	15	9
IV	21	9	12	4
V	42	19	21	16
VI	35	14	18	13
VII	38	14	20	15

## Data Availability

Not applicable.

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
