# Peer review of "Genome-Wide Identification and Expression Analysis of nsLTP Gene Family in Rapeseed (Brassica napus) Reveals Their Critical Roles in Biotic and Abiotic Stress Responses"

_ijms, 2022, doi:10.3390/ijms23158372_

Round 1
Reviewer 1 Report
Most of the revision was completed. Some of revision rest are as follows.
Line 223: What is the organ of rapeseed that used in Fig. 5? Figure 5. Expression patterns of BnLTP family genes in cotyledons of? susceptible (Westar) and tolerant (ZY821) genotypes of rapeseed~
Lines 40 to 245: Table S21 showed the results of IAA expression pattern. Could you show the IAA results in Figure 7, please? Because contents of the lines mentioned mainly IAA and ABA.
Figure 9: Move to Fig. 9 to upper part of Discussion, please.

Author Response
Thanks a lot for your positive and constructive comments and suggestions.
Most of the revision was completed. Some of revision rest are as follows.
Line 223: What is the organ of rapeseed that used in Fig. 5? Figure 5. Expression patterns of BnLTP family genes in cotyledons of? susceptible (Westar) and tolerant (ZY821) genotypes of rapeseed~
Reply: The organs of rapeseed in Figure 5 are leaves, and we have modified it.
Lines 40 to 245: Table S21 showed the results of IAA expression pattern. Could you show the IAA results in Figure 7, please? Because contents of the lines mentioned mainly IAA and ABA.
Reply: We could not show the IAA results in Figure 7, because RNA-seq data of IAA and ABA are come from different NCBI GEO dataset and thus their dada cannot be normalized and compared together.
Figure 9: Move to Fig. 9 to upper part of Discussion, please.
Reply: We have Move to Figure 9 (now is Figure 10 in new revised manuscript) to upper part of Discussion.
Reviewer 2 Report
see attachment

Author Response
Thank you very much for your positive and constructive comments and suggestions.
I did notice a few additional places that corrections could be made:
Line 39: study of the genetic;
Line 53: nsLTPs is of:
Line 54: with enhanced seed;
Line 67: there has been no systematic whole-genome:
Line 84: is absent in;
Line 99: as shown in;
Line 188: Transcriptome data [in total would be ok, but you use it in the next paragraph, so it
seems ok to omit this phrase here];
Line 193: the embryo-specific gene;
Line 214: transcriptome data of the;
Line 255: the numbers of;
Line 256: also increased gradually.
Reply: We have modified and revised the above all corrections, and also check the whole manuscript and revived all the problems newly found.
This manuscript is a resubmission of an earlier submission. The following is a list of the peer review reports and author responses from that submission.
Round 1
Reviewer 1 Report
The paper by Xue et al., on the complex nature of nsLTPs in Brassica highlights the benefits that can be derived from untiring genome mining. They have sifted through as many datasets as they were able to find to document the gene structure, genomic location, and expression profile of each member of this very large family. They have, in addition, incorporated an overwhelming number of case histories and examples to illustrate the importance of specific members of this group in other species.
The writing is, for the most part, acceptable, but nevertheless, could benefit from improvements. As it stands, some readers would be unsettled (like riding a bike on a rocky road-unsettled) by the errors involving the absence of definite and indefinite articles (the and a/an), and sometimes the failure to employ singular and plural words. I have tried to note a few of these instances in my comments below, but I am certain I did not exhaust the problem.
Before getting to those editorial suggestions, I do want to note 2 other points:
1. Although I also like to cite more than the average number of references, in this case, I feel the authors made an attempt to be “overly encyclopedic”, to the point I found myself skipping whole paragraphs to get to a new point. Although they did nothing wrong by being so thorough, I personally think it may be counterproductive to write a research article with the same excruciating detail that one would write a review. In relation to my next point, despite providing us with a broad review of nsLTPs, when the authors cite evidence that these proteins do in fact bind lipids, they first reference 2 reviews and 1 research paper (Line 46) and later (Line 375), 5 more papers that (and I apologize if I am wrong about this) appear to be reviews based on their titles. Perhaps the authors could clarify how many of these varied proteins have been experimentally shown to bind lipids under biological conditions (and if the information is available, how many failed to do so).
2. I am concerned about what they define as a nsLTP. When I look at Figure S1, I can see that many proteins share no common domains. The authors themselves noted that the size of these proteins can vary from 40 to 400 amino acids (line 164). Some are constitutive, some induced, some induced by pathogens, some by specific abiotic stresses. I cannot argue that these proteins may have evolved from a common ancestor, but I question whether they are functionally related to each other today. Couldn’t the diagnostic motif 8CM motif (line 165) be nothing more than a basic building block, like an ATP-binding, or a nuclear targeting, domain? I am concerned that by lumping so many unrelated proteins together, the authors are blurring the differences that exist between them, both structurally and functionally. Imagine, for example, a paper on all proteins with nuclear targeting domains (transcription factors, DNA polymerases, splicing enzymes, E3 ligases, DNA methyltransferases, etc.). Would lumping those into one family reveal much of their function, or would we recognize that each must be dealt with on its own? My feeling is that they should focus on one branch of similarly structured proteins and try to understand/explain how they are (or are not) functionally similar.
Comments on the style (underlines indicate things that could be added):
1. Fig. S1: This figure and others really show say where the reader can go to find the sequence of the domains indicated as motifs 1-20.
2. I am not sure if the journal is satisfied with the number of supplementary figures and tables, but found hunting through them (and expanding the font), tedious. This might be helped if the authors restricted themselves to 1 class of this family.
Line 13: the nsLTP gene;
Lines 14-15: distributed randomly among 19 chromosomes of rapeseed.;
Line 25: functional study;
Line 31: crops;
Line 35: suffer from extensive losses;
Line 46: have been shown;
Line 49: also play critical;
Line 50: plant responses to;
Line 52: a systematic;
Line 55: delete the first “plant” (it is repetitive if you then say higher plants a few words later);
Line 75: functional;
Line 104: my fault, but I am not sure what logo means in this sentence;
Line 107: the GFF3;
Line 108: in the B.;
Line 118: in an evolutionary;
Line 119: I’m not sure what you want here. Maybe “which were also annotated as a duplicated…”?;
Line 123: from the B.;
Line 130: 42 different tissues:
Line 146: of a Sclerotinia;
Line 148: an RNAsimple;
Line 158: fluorescence;
Lines 169-170: thus they all could be substrates for the cell’s secretory pathway:
Line 176: y-axis shows;
Line 193: from a B.;
Line 197: used to perform;
Line 217: played an important;
Line 222-223: might have resulted from;
Line 233: in the first type;
Line 234: included sequences associated with the differentiation;
Line 235-243: delete the repetitive “element” and just state the trait. You already said in 234 that these are examples of cis elements;
Line 250: seriously, was any kind of element not included in this list? It seemed to encompass all that I know about plants;
Line 276: in the embryo proper;
Line 313: I don’t know how to interpret this. Each kind of treatment can reveal different sets of genes depending on time of treatment, and magnitude of treatment (e.g., the concentration of hormones). I would be hard pressed to use your data figure to predict how my own gene would respond without knowing the actual parameters of the treatment applied in your case. Was every treatment applied for the same time, were the amounts of hormones, stresses, etc. biologically meaningful, etc, or excessive in order to ensure they would have an effect? I also found the way the paper listed numbers (e.g., 65, 74, 76,80, and 67; 12, 26, 61, 41, and 32 respectively) opaque. I assume these are the number of genes responding to the treatments listed elsewhere in the paragraph, but why not write a complete sentence that says that?;
Line 348: does the upregulation of a gene automatically link it to the color, or could it just be co-regulated? Obviously one helpful experiment would be to knock out the gene and show the seed color changes. Did you do this?
Line 355-360: I think there was a formatting error here. Perhaps line 355 should be part of the same sentence as 356? You also might correct the spelling of fluorescence.
Line 387: might have contributed;
Line 427: studies;
Line 442: I would replace Substantial with Numerous;
Line 449-450: I think you mean mutants, not mutations (?);
Line 453: BnLTP085, displayed seed-specific expression, revealing;
Line 459: you can only attribute high germination numbers to the expression of the genes if you know the lines are otherwise isogenic;
Line 463: delete mainly;
Line 467: further functional research.
Reviewer 2 Report
The research focused on;
comprehensive whole-genome analysis of BnLTP family genes in rapeseed, including phylogenetic relationship, exon/intron structure, MEME motifs, synteny analysis, SSR loci, promoter cis-elements and the gene expression patterns. The research showed roles of BnLTP family genes in plant growth and development and in response to biotic/abiotic stresses and phytohormone treatments.
Some of questions are rest. Please, consult on your manuscript with detailed revision.
*Materials and Methods
-Lines 129 to 144: Treatment conditions (time intervals) will be needed at the expression analyses of biotic stress, abiotic stress and germination.
*Minor revision was checked on the manuscript paper.
